# Problems, Stressors and Needs of Children and Adolescents with Cancer

**DOI:** 10.3390/children8121173

**Published:** 2021-12-10

**Authors:** Anna Lewandowska, Barbara Zych, Katalin Papp, Dana Zrubcová, Helena Kadučáková, Mária Šupínová, Serap Ejder Apay, Małgorzata Nagórska

**Affiliations:** 1Institute of Healthcare, State School of Technology and Economics, 37-500 Jaroslaw, Poland; 2Institute of Health Sciences, Medical College of Rzeszow University, 35-310 Rzeszow, Poland; ba.zyc@wp.pl; 3Faculty of Health, University of Debrecen, 4400 Nyíregyháza, Hungary; papp.katalin@foh.unideb.hu; 4Faculty of Social Sciences and Health Care, Constantine the Philosopher University in Nitra, 94974 Nitra, Slovakia; dzrubcova@ukf.sk; 5Faculty of Health, Catholic University in Ružomberok, 03401 Ružomberok, Slovakia; helena.kaducakova@ku.sk (H.K.); maria.supinova@szu.sk (M.Š.); 6Department of Midwifery, Faculty of Health Science, Ataturk University, Erzurum 25240, Turkey; sejder@atauni.edu.tr; 7Institute of Medical Sciences, Medical College of Rzeszow University, 35-310 Rzeszow, Poland; ma.nagorska@gmail.com

**Keywords:** cancer, children, adolescents, problems, stressors, needs

## Abstract

Background: Cancer diseases in children and adolescents are considered to be one of the most serious health problems in the world. It is estimated that about 151,435 cases are diagnosed in children annually. Children with cancer experience many comorbid symptoms related to diagnosis and treatment that can profoundly affect their lives. They experience physical and emotional suffering, which affects their well-being and physical fitness, influencing the prognosis and deteriorating their physical, mental and social functioning. Given the limited data, an attempt was made to assess the problems of the biopsychosocial sphere of need and stressors among children and adolescents treated for cancer. Accurate symptom assessment is essential to ensure high-quality care and effective treatment. Patients and Methods: The qualitative study was conducted in pediatric oncology of hospitals in Poland. Children diagnosed with cancer were invited to participate in the study to assess their problems, stressors and needs. Results: The study included 520 people, where female sex constituted 48% and male 52%. The mean age of the children is 13.2 SD = 2.5. Negative experiences related to the disease are experienced by 82% of children. Among the surveyed children, the most experienced were anxiety (61%). The conducted research shows that as many as 69% of all respondents experienced states that indicate severe depression. The most common somatic problems reported by children were pain (58%). The most dominant areas of life that had a negative impact was body image (85%). Conclusions: Children and adolescents diagnosed with neoplastic disease experience many problems and stressors in every sphere of life, which undoubtedly affects a high level of unmet needs. The main category of needs concerning the challenges faced by children with cancer was psychological and care problems. In the youth group, the needs were mainly related to education and social support.

## 1. Introduction

Cancer diseases in children and adolescents are considered to be one of the most serious health problems in the world. It is estimated that about 151,435 cases are diagnosed in children annually [1,2]. According to data from the World Health Organization, approximately 400,000 children and adolescents aged 0–19 are diagnosed with cancer each year. The incidence of cancer in developed countries ranges from 110 to 150 children per million, which means that 1 per 600 to 1 per 450 children will develop cancer during the first 15 years of life. In high-income countries where comprehensive services are generally available, more than 80% of children with cancer are cured. It is estimated that in low- and middle-income countries, 15–45% are cured [3,4,5]. In the United States, after accidents, cancer is the second leading cause of death in children aged 1 to 14. In 2021, around 10,500 children under the age of 15 are expected to be diagnosed with cancer, and around 1190 children under the age of 15 will die from cancer [6]. In the European Region, childhood cancers account for 1–1.5% of cancers in the total population, with more than 3 million new cases of cancer and 1.7 million deaths each year [1,2,3,4,5]. In Europe, there are approximately 15,000 new cancer cases every year among children from 0 to 14 years of age, and among adolescents and young adults aged 15–24 years, another 20,000 cases. In Poland, 1100–1200 new cases occur per year, and the incidence rate is close to 140–145 new cases per 1 million children and adolescents. In Poland, there are approximately 8 million children and adolescents aged 0 to 17, and approximately 10,000 children received treatment for cancer. In the past decades, the incidence in children has been increasing in both sexes. The average annual rate of incidence increase is about 1.2% per year [7,8,9,10,11,12].

Children with cancer experience many comorbid symptoms related to diagnosis and treatment that can profoundly affect their lives. They experience physical and emotional suffering, which affects their well-being and physical fitness, leading to delays or limitation of treatment, influencing the prognosis and deteriorating their physical, mental and social functioning [13,14,15]. Pediatric oncology shows that stressors related to the disease occur more frequently and more intensely, causing higher anxiety and lower self-esteem. The impact of the diagnosis and experience of cancer in a child may be extreme and permanent, causing crises, also affecting life in adulthood [16,17,18]. During the disease, new needs emerge, which, if not satisfied, increase the negative effects of neoplastic disease, including the fear of death, depression or reduced pain tolerance [14,19].

Although cancer is a common problem, the knowledge about the experiences of Polish children and adolescents treated for cancer is very limited. Given the limited data, an attempt was made to assess the problems of the biopsychosocial sphere of need and stressors among children and adolescents treated for cancer. Accurate symptom assessment is essential to ensure high-quality care and effective treatment.

## 2. Objective of the Research

The study aims to analyze the problems, stressors and needs of children and adolescents hospitalized due to cancer, which gives the opportunity to better understand the experiences of children, and for health care professionals, it allows a broader view of a child with cancer.

## 3. Materials and Methods

### 3.1. Study Design

The qualitative study was conducted in pediatric oncology and hematology departments of hospitals in Podkarpackie Province in 2009–2019. Hospitals provided treatment and care by the public health care system to pediatric patients diagnosed with cancer. Children diagnosed with cancer were invited to participate in the study to assess their problems, stressors and needs. Due to the small size of the sample, the share of respondents with fairly consistent characteristics was important. The size of the sample resulted from the nature of the study (preliminary study).

Each invited person and their guardian were informed about the purpose of the study. The eligible patient and guardian received an information pack from a research group member. The information package consisted of a letter describing the objectives of the study and its course, a consent form to participate in the study, to be completed if patients are interested in the study, and a no consent sheet if they were not. After informed consent was given by the patient and their caregiver, the patient chose the time and place to participate in a face-to-face interview conducted in the clinic by a research group member. The interview lasted approximately 60 min. In case of the patient’s fatigue, the interview was divided into parts to maintain their physical and mental comfort. All participants were asked the same questions, and the interviewer was able to explain the content of the questions if necessary. However, the questions did not seem difficult for any participant, including younger ones, and no explanations were needed.

### 3.2. Participant Recruitment, Inclusion and Exclusion Criteria

The inclusion criteria were the confirmed diagnosis of childhood cancer in the pathological report, age from 10 to 18 years, no previous chronic or life-threatening disease, knowledge of the Polish language and the consent and willingness of parents to participate in the study. Eligible participants were randomly selected from the general population. The main indicators of participation in the study were the cancer diagnosis at least three months before the study, life expectancy > 6 months, age and awareness of the diagnosis.

The exclusion criterion was the diagnosis of neoplastic disease shorter than three months because the initial period of diagnosis and treatment is associated with an enormous psychological burden and the need to adapt to the patient’s situation, which may introduce errors in the results. The study excluded patients with cognitive impairment, patients who did not express their willingness to participate in the study, those under that could not speak Polish.

### 3.3. Participants

Children and adolescents were invited to participate in the study during their hospitalization. Each invited person was informed about the purpose of the study. After obtaining the written, informed consent of the child and the guardian, the subject was interviewed. The participants chose the place and time of the conversation and decided whether they wanted the parent’s participation in the conversation. The non-participation rate was 10%. Parents of children who chose not to participate reported that they did not have enough time or that their children were tired or not feeling well. Non-participation data are not available.

### 3.4. Research Procedures

The study was approved by the Bioethics Committee (Resolution No. 386/2009 and 4 December 2017). Participation in this study was voluntary and anonymous, and respondents were informed of their right to refuse or withdraw from the study at any time. Each participant was informed about the purpose of the study and the time of completion of the study. Families were invited to participate during the child’s hospitalization or an outpatient visit. After giving informed consent, the parents completed the questionnaire.

### 3.5. Method

#### 3.5.1. Clinical Interview

The method used in the research was a clinical, direct, individual, structured interview, which was in-depth and focused. The qualitative interview was a standardized measuring instrument, verified by testing a group of 20 patients during the month. It contained open-ended, single and multiple-choice questions, allowing to obtain demographic and epidemiological information concerning problems related to the disease: daily stressors of the child, somatic, psychological and social problems and needs.

#### 3.5.2. Visual Analogue Scale (VAS)

It is a reliable tool for determining the severity of pain. Cyclically repeated measurements of pain intensity using the VAS scale enable the assessment of the effectiveness of analgesic treatment. The scale is a 10 cm ruler, where 0 is no pain at all, 1–3 is mild, 4–6 is moderate and 7–10 is severe.

#### 3.5.3. Beck Questionnaire for Children to Assess Emotional and Social Disorders (BYI)

This questionnaire is used as an aid in individual diagnostics, serving to objectify and quantify the degree of disturbances in the child’s functioning. It is a tool covering a wide spectrum of disorders, intended for filling by schoolchildren. It can be used in situations where it is not possible to obtain information from parents or serve as a supplement to them, constituting an important source of information on the child’s functioning. Having a high score for your child can help highlight symptoms that would otherwise go unnoticed and allow them to be referred to a specialist. The scale consists of 21 multiple-choice questions, in which one should give one of four possible answers, and then summarize the obtained points and compare them to the norms: no depression (0–11 points), mild depression (12–19 points), moderate depression (20–25 points) and severe depression (26–63 points).

### 3.6. Data Analysis

The analysis used descriptive statistics and confidence intervals in the assessment of participants’ characteristics, metric and demographic data, and in the assessment of problems. Statistical characteristics of continuous variables are presented in the form of arithmetic means, standard deviations and medians. Statistical characteristics of step and qualitative variables were presented in the form of numerical and percentage distributions, using the Student *t*-test or the Mann–Whitney U test. Correlations were determined using Pearson’s test, while χ^2^ was used for intergroup comparison. Significance was assessed at the level of *p* < 0.05. The manuscript presents the results regarding the comparison of arithmetic means and standard deviation obtained on the Beck scale. The non-parametric t-Student test was used for the comparative analysis. Following the applicable statistical principles, raw results were used for intergroup comparisons. The repeatability of answers to individual questions was assessed using Kappa Cohen statistics.

## 4. Results

### 4.1. Demographic Data

The study included 520 people, where female sex constituted 48% and male 52%. The mean age of the children is 13.2 SD = 2.5. Hospitalized for the first time constituted 52% of all respondents, once again 48%. The data analysis shows that the most common neoplasms in the study group are leukemias (44%). All children had a caregiver; most were mothers (83%), while the remaining 17% were fathers. All participants received chemotherapy (100%). In total, 25% of the respondents were primary school students, 41% were middle school students and 34% were high school students. Other descriptive statistics identifying the studied group are presented in Table 1.

### 4.2. Psychological Problems

Negative experiences related to the disease are experienced by 82% (95% CI: 79–85) of children. Among the surveyed children, the most experienced were anxiety (61%, 95% CI: 60–63), depression (58%, 95% CI: 55–60) and anger (33%, 95% CI: 31–35) (Figure 1). As many as 42% (95% CI: 39–45) of children could not come to terms with the disease, 28% (95% CI: 27–30) with hospitalization, 20% (95% CI: 19–22) with homesickness, 16% (95% CI: 14–18) with parents’ emotions and suffering, 14% (95% CI: 10–18) with unjust fate and 24% (95% CI: 19–28) with a lack of physical activity. The factors negatively influencing well-being mentioned by patients were most often isolation (76%, 95% CI: 75–78) (Figure 2). Subsequently, the intensification of features indicating the existence of depression in children was examined. Comparison of the results obtained on the Beck scale in groups of girls and boys shows statistically significant differences (*p* < 0.01). Girls experienced emotional states indicative of depression relatively more often than boys. It was necessary to convert the raw results obtained on the Beck scale according to the norms to determine the number of people affected by the problem of depression. The conducted research shows that as many as 69% of all respondents (95% CI: 63–72) experienced states that indicate severe depression (Table 2). Then, the differences in the children’s depression in terms of hospitalization time were determined. A comparison of the arithmetic means and standard deviation obtained on the Beck Depression Scale showed no significant differences (*p* < 0.94). This means that the severity of depression features among children with the comparison of hospitalization time is similar (Table 3).

### 4.3. Communication Problems

All examined children were aware of the disease (100%). The vast majority of them received information about the disease from their parents (69%, 95% CI: 67–71), while the remaining children received such information from health care professionals (31%, 95% CI: 30–32). When assessing the impact of the disease on the child’s contacts with family and friends, it was shown that children most often complained of communication problems due to isolation (79%, 95% CI: 77–81), shame about appearance (67%, 95% CI: 65–70) and overprotection (35%, 95% CI: 33–37). Most parents spent a great deal of time with their children visiting them daily (82%, 95% CI: 80–84); however, 88% (95% CI: 87–89) of children would like to spend more time with their family. According to 64% (95% CI: 62–66) of the respondents, parents had not changed their attitude towards them since the diagnosis of the disease, 28% (95% CI: 27–29) believed that they had become more caring and 8% (95% CI: 7–9) that they had more time for them. The changes in parents’ behavior noticed by the children were mainly sadness (54%, 95% CI: 52–58) (Figure 3). A total of 76% (95% CI: 74–79) of patients maintained contact with their peers less than once a week, while only 6% (95% CI: 5–8) were visited by their peers every day. Only 20% (95% CI: 17–23) attended hospital school, 62% (95% CI: 60–65) were educated with parental support and 18% (95% CI: 15–20) were not educated.

### 4.4. Somatic Problems

The most common somatic problems reported by children were pain (58%, 95% CI: 55–60), weakness (51%, 95% CI: 48–55) and vomiting (39%, 95% CI: 35–42) (Table 4). In assessing pain intensity on the VAS scale, the mean was 57% (95% CI: 54–59). The proportion of patients assessing their mean pain in the last week as mild was 22% (95% CI: 19–25), moderate 48% (95% CI: 45–49) and severe 30% (95% CI: 28–32). The children most frequently indicated that pain peaked in the morning and decreased over the day (88%, 95% CI: 85–91). Younger children experienced breakthrough pain more often than adolescents (73% vs. 33%, *p* ≤ 0.01). Gender was not significantly correlated, as was the incidence of breakthrough pain with anxiety or depression. Children usually described the pain as acute (79%, 95% CI: 75–81).

### 4.5. Caring Problems

Factors that, in the opinion of children and adolescents, adversely affected their stay in the ward included limitations in going outside (68%, 95% CI: 67–69), multiple procedures (40%, 95% CI: 39–41), no intimacy (36%, 95% CI: 34–39) as well as unfriendly staff (6%, 95% CI: 4–9). Despite the presence of factors hindering hospitalization, patients were also able to perceive amenities. The most popular entertainment preferred by respondents is watching TV (56%, 95% CI: 54–59) and computer games (48%, 95% CI: 45–49) (Figure 4). The study also attempted to assess the impact of hospital staff on the hospitalization process. Nursing staff assessed 46% (95% CI: 45–49) children very well, medical staff rated 14% (95% CI: 12–18) patients very well, while 40% (95% CI: 38–44) negatively assessed the staff. In total, 80% (95% CI: 79–81) of patients did not confide in health professionals and only 20% (95% CI: 18–22) talked to workers about illness and anxiety.

### 4.6. Stressors

In the study, children and adolescents were asked about the stressors that are of the greatest importance to them. With little differences between the age groups, the most dominant areas of life that had a negative impact were body image (85%, 95% CI: 84–86), life control (57%, 95% CI: 55–58) and plans for the future (27%, 95% CI: 24–29). Significant differences in the negative impact items for the younger age group compared to the older age group concerned education plans (17%: vs. 38% *p* < 0.01) and peer relationships (44% vs. 58%; *p* < 0.01). Young people were significantly more worried about their parents’ finances (*p* = 0.01) (Table 5).

### 4.7. Needs

The most common unmet needs were psychological needs, care needs and social needs (Table 6). Age was associated with a higher level of unmet needs in the fields of psychology, health care and support. Children consistently showed a higher level of unmet needs than adolescents (*p* = 0.01).

## 5. Discussion

Our study is one of the few that identifies problems and needs among children treated for cancer in Poland. This study focused on somatic, emotional and social problems and needs resulting from illness and hospitalization.

The first thread of this study showed that children had numerous psychological problems resulting from the disease. In total, 82% of children have negative experiences related to the disease; most often, this was fear (61%), depression (58%) and anger (33%). The factor negatively influencing wellbeing mentioned by patients was most often isolation (76%). Similar results were obtained by Reisi-Dehkordi et al., demonstrating that children had psychological problems in contact with the disease; it was found that depression, anxiety and aggression were common among children with cancer [20]. Additionally, the study by Kohi et.al showed that the children were emotionally affected, felt separated from family members and had negative feelings related to the diagnosis [21], which was also confirmed by other studies [22,23]. The results presented by Dyson et al. also showed that only a few participants (17%) presented a lower level of distress and unmet psychological needs [24]. Similarly to our studies, where 69% of the respondents experienced conditions indicative of severe depression in the past, studies by Hedström and others showed that 12% of children reached the cut-off point for potential clinical anxiety and 21% for potential clinical depression. Mental health and vitality scores were lower than the normative values [25]. The problem of depression is generally more common in girls than boys, which our research also confirms. Girls experienced emotional states indicative of depression relatively more often than boys (*p* < 0.01). This is true of both so-called severe and moderate depression. According to the classification of mental and behavioral disorders, severe depression is linked to feelings of worthlessness, low self-esteem, guilt and suicidal thoughts. Moderate depression is characterized by the loss of interest and pleasure, as well as increased fatigue [26]. What is very worrying among people who obtained a result in the clinical scope of depression, a higher frequency of pain associated with procedures, treatment [25], sleep, appetite, concentration disorders and fatigue were more frequent [25,26]. The results of a study by Verberne et al. showed that children with cancer developed insomnia, causing fatigue and deterioration of their functions [27].

Another element of this study showed that children had communication problems in contact with the disease. What hinders communication are isolation (79%) and shame about appearance (67%), as well as changes in parents’ behavior noticed by children. The vast majority of patients maintained contact with their peers less than once a week. Our results are in line with the research of Lee et al., where patients encountered interpersonal communication problems due to a disease affecting their body image [28]. Kamper et al. claimed that peers paid special attention to the patient due to their illness, which hindered their interpersonal communication [29]; in the study by Reisi-Dehkordi et al., peers isolated patients due to the disease [20].

Somatic problems, which are complex and change over time, are a major challenge in pediatric oncology. Pain and treatment side effects, especially nausea, are a constant problem for many children and adolescents with cancer, as confirmed by numerous studies [22,30,31,32,33,34,35]. Regardless of whether the pain is associated with the disease, treatment or procedures, it is the most frequently identified and effectively treated symptom in children and adolescents [36]. Van Cleve et al. showed that the highest mean pain intensity scores in children with acute lymphoblastic leukemia were directly related to disease development and treatment or diagnostic procedures. The children indicated that the most common places of pain were legs, abdomen, head, neck and back [37]. In our research, assessing pain intensity on the VAS scale, the mean was 57%. The proportion of patients assessing their mean pain in the last week as mild was 22%, moderate 48% and severe 30%. The children most frequently indicated that pain peaked in the morning and decreased over the day (88%). Fatigue and excessive sleepiness seem to be equally significant problems for cancer patients [32,33,38,39,40,41], as well as the loss of the ability to do things as before (e.g., playing sports, playing and studying) [22,39]. The study of potential factors contributing to fatigue, such as sleep disturbances, was made a priority in childhood cancer research. The main aim of the study by Zupanec et al. was to investigate the relationship between sleep habits, sleep disturbances and fatigue in children receiving maintenance chemotherapy. Sleep disturbances were common in children (87%) and the results of sleep disturbances were positively correlated with the assessment of fatigue [38]. A study by Hedström et al. showed that in children 12 years of age or younger, the most frequently mentioned and most disturbing symptoms were pain associated with diagnostic procedures and treatment, nausea and fatigue while in children 13 years of age or older, the most bothersome symptom was nausea. In adolescents, physical symptoms and emotional stress were related [42]. In another study, it was shown that for most adolescents, the greatest concerns were related to the deterioration of health, mucositis, nausea, pain caused by procedures and treatments and leaving school [43]. In the publication by Enskär and von Essen, it was found that two-thirds of the children tested experienced physical symptoms. Children treated for cancer significantly more often reported worries about hair loss (47% vs. 10%) and nausea (47% vs. 14%) (*p* < 0.05 for each) and fatigue (65% vs. 43%) compared to children who ended treatment [44]. Other authors found that physical suffering, defined as fatigue, eating problems, hair loss and problems taking medications, was reported by two-thirds of adolescents undergoing cancer treatment. More than half of the adolescents who completed treatment reported that fatigue and eating problems continued, and that life was less fulfilling [45,46]. The above considerations confirm our results, where the most frequently reported symptoms by children were pain (58%), weakness (51%) and vomiting (39%).

Another piece of the study showed that children who were exposed to the disease faced care problems. Factors that, in the opinion of children, adversely affected their stay in the ward were mainly limitations in going outside (68%) and the multiplicity of procedures (40%). Despite the presence of factors hindering hospitalization, patients were also able to perceive amenities. The most popular entertainment preferred by respondents is watching TV (56%) and computer games (48%). In studies by Chien et al., participants expressed concerns about poor oncology care associated with misdiagnosis, as well as treatment leading to multiple visits to healthcare before a correct diagnosis [22]. Similar problems were presented in the Edwards and Greeff studies, which identified challenges with poor services, including the need to travel long distances for treatment, poor care in hospitals and delays in cancer diagnosis [47]. No other studies were found for the subcategory of care problems in children and adolescents.

Studies involving children and adolescents with cancer who are cancer survivors clearly show that cancer and its treatment can affect various spheres of life [48]. However, little is known about the stressors that children feel are of the greatest importance to them. One of the first studies in this area was carried out by McCaffrey, in which children spoke about cancer-related stressors in individual interviews. The main stressors were medication and procedures, loss of body image control, fear of death, alopecia and the inability to be with friends. Experiencing these stressors resulted in low self-esteem, feelings of unhappiness or fatigue, becoming more mature than their friends, dropping out of school and not participating in sports and weight loss [49]. In another study, one of the more frequently reported stressors was poor body image, which may lead to low self-esteem and affect the ability to form healthy peer relationships [50]. In the study of body image and social adaptation in adolescents with cancer and healthy controls by Pendley et al., body image was assessed more negatively in cancer patients, and this negative assessment was more pronounced for a longer time after treatment [51]. In the studies by Bellizzi et al., the most frequently reported negative life domains in adolescents with cancer were the financial situation, body appearance, sense of control over life and disruption of education [52]. The pediatric literature suggests that time spent outside of school as a result of treatment may have severe effects, such as depression, poor self-esteem and lack of interest, as well as longer-term effects, such as loss of purpose or difficulty re-engaging in educational activities [53]. As a result of limited access to other studies, especially in recent years, we have attempted to identify stressors for adolescents. The stressors that matter most to them are the negative impact of the disease on their body image (85%), a negative impact on their plans for the future (27%), and the lack of control over their lives (57%). Our results confirm previous reports, and what is most interesting, we showed significant differences among age groups.

In the last part of the study, the presence of needs among the surveyed children was assessed. This is a particularly important issue as the unmet psychosocial needs of children and adolescents with cancer increase the negative effects of cancer, fear of death, depression, reduced pain tolerance and disability [18]. This may lead to unnecessary, repeated, frequent hospitalizations or invasive procedures and lowered quality of life [54]. According to our results, children and adolescents most often reported psychological needs, care needs, as well as physical and life needs. Kohi et al. identified needs which included the need for improved hospital care by staff, the need for community support, financial needs, the need for better cancer care and treatment in hospitals and the need for increased cancer education [21]. Research by Bonevski and others shows that the most common needs are to provide more information about the diagnosis and prognosis (63%), to improve communication with doctors (50%), increase intimacy in doctor’s offices (85%) and improve the organization of care (76%) [55]. On the other hand, Dyson et al. showed that the most often unmet needs were physical and life needs, followed by psychological needs, needs related to the healthcare system and information and needs related to care and support [24]. What our research has shown is that age was associated with a higher level of unmet needs in the fields of psychology, health care and support.

Our qualitative study includes many predictors of cancer in children and adolescents. The problems, stressors or needs of cancer treatment usually do not occur in isolation, but often occur as multiple problems simultaneously. Our research is multiplicative, which probably supplements the existing gaps in the literature. Most of the research on children remains descriptive. Intervention studies that focus on the problems identified are lacking. In addition, the strength of the study is the confirmation of the results of the studies described in the literature, and the obtained results provide better insight into the analysis of the problems of children and adolescents in all areas of life.

The limiting factors are the small size of the research sample and its heterogeneity in terms of diagnosis. Access to the sample is difficult, so we are planning multi-center studies in the future to help develop larger datasets on age groups. We are in the process of reporting studies from all over Poland to other countries.

## 6. Conclusions

Children and adolescents diagnosed with neoplastic disease experience many problems and stressors in every sphere of life, which undoubtedly affects the high level of unmet needs. The main category of needs concerning the challenges faced by children with cancer was psychological and care problems. In the youth group, the needs were mainly related to education and social support.A thorough symptom assessment at an early stage and during treatment is essential in caring for the development of early and supportive interventions for children and adolescents with cancer.

## Figures and Tables

**Figure 1 children-08-01173-f001:**
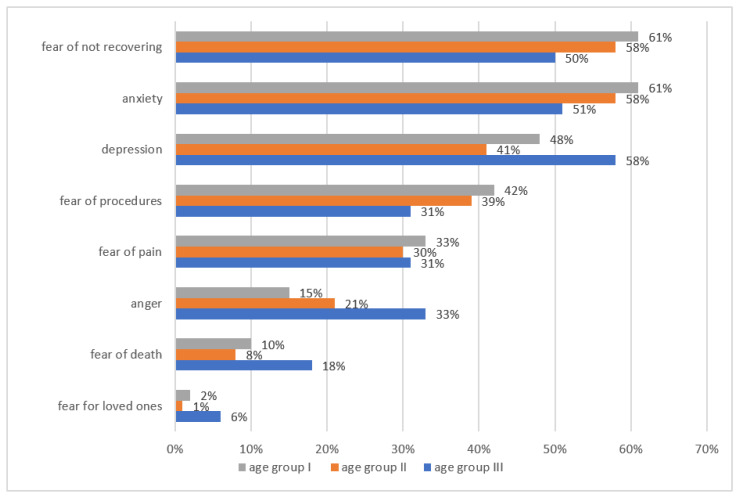
Negative experiences related to the disease among the respondents.

**Figure 2 children-08-01173-f002:**
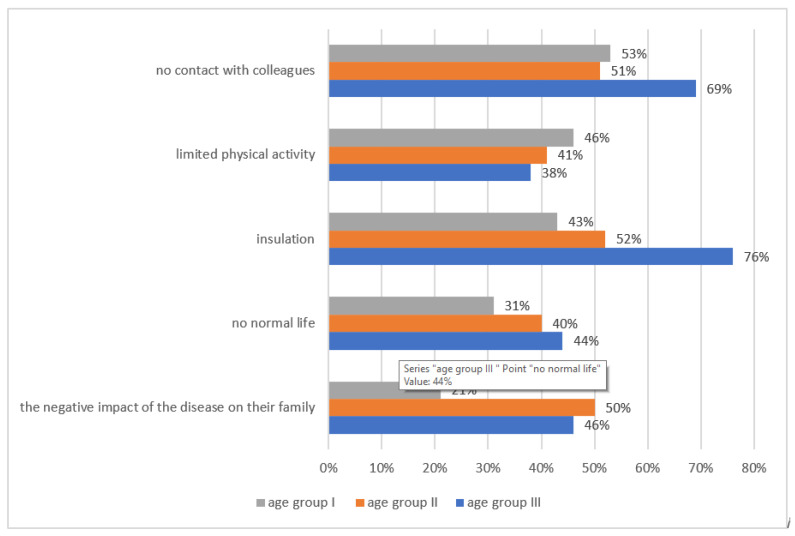
Factors negatively influencing the well-being of the respondents.

**Figure 3 children-08-01173-f003:**
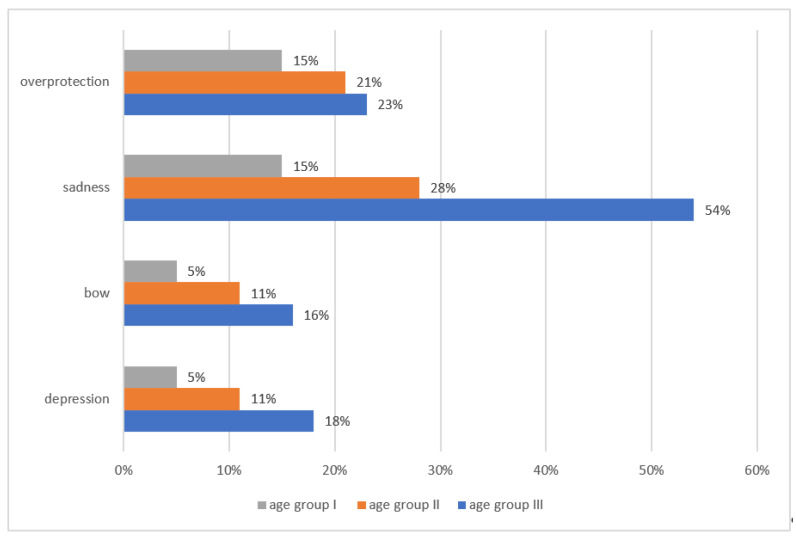
Changes in parents’ behavior noticed by the respondents.

**Figure 4 children-08-01173-f004:**
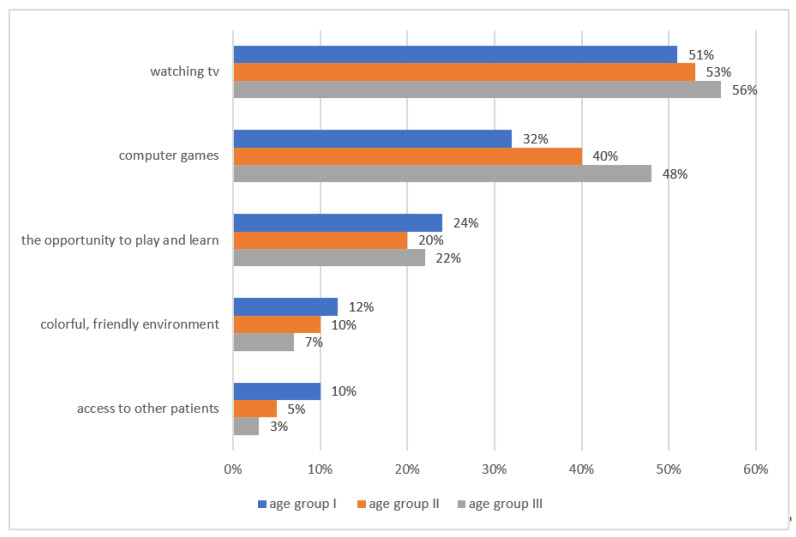
Hospital facilities appreciated by respondents.

**Table 1 children-08-01173-t001:** Descriptive statistics of the examined group of patients.

Demographic Information	Total *n* = 520	*p*
Characteristics% (*n*)		
Sex		
girls	48% (250)	0.19
boys	52% (270)	
The age of the study group		
SD	13.2 (2.51)	
95%CI	<10; 18>	
Place of residence		
city	59% (307)	0.21
village	41% (213)	
Age groups		
age group I 10–12 years	31% (161)	0.12
age group II 13–15 years	35% (182)	
age group III 16–18 years	34% (177)	
Education of the study group		
primary school	25% (130)	0.19
junior high school	41% (213)	
high school	34% (177)	
Type of cancer		
leukemia	44% (229)	0.07
lymphomas	18% (93)	
Ewing’s sarcoma	14% (73)	
osteosarcoma	14% (73)	
nephroblastoma	10% (52%)	
Family status		
full family	79% (411)	0.01
incomplete family	21% (109)	
Times of illness		
3–12 months	43% (224)	0.01
1–2 years	47% (244)	
3–4 years	10% (52)	

**Table 2 children-08-01173-t002:** Severity of depression on the Beck scale.

Factors	Girls		Boys		t	*p*
	x¯	s	x¯	s		
Beck scores	32.2	9.4	27.7	8.6	4.2	0.01

**Table 3 children-08-01173-t003:** Severity of depression on the Beck scale.

Factors	First Hospitalization		Second Hospitalization		t	*p*
	x¯	s	x¯	s		
Beck scores	29.9	10.1	30.2	11.1	0.20	0.94

**Table 4 children-08-01173-t004:** Somatic symptoms among respondents.

Symptoms	Age			*p*	Sex		*p*
	10–12	13–15	16–18		Girls	Boys	
Characteristics *n*/%							
Pain	60% (97)	55% (100)	21% (37)	0.01	60% (150)	55% (148)	0.91
Nausea	13% (21)	10% (18)	10% (18)	0.41	13% (32)	10% (27)	0.55
Vomiting	56% (90)	39% (71)	19% (34)	0.01	56% (140)	42% (113)	0.44
Weakness	56% (90)	46% (84)	27% (48)	0.77	56% (140)	42% (113)	0.55
Somnolence	28% (45)	27% (49)	11% (19)	0.91	30% (75)	27% (73)	0.91
Fatigue	48% (77)	37% (67)	31% (55)	0.54	48% (120)	37% (100)	0.91
Difficulty sleeping	23% (37)	30% (55)	30% (53)	0.55	23% (57)	30% (81)	0.71
Difficulty concentrating	16% (26)	30% (55)	44% (78)	0.01	46% (115)	30% (81)	0.74
Loss of the ability to do things before getting sick	38% (61)	17% (31)	31% (55)	0.41	40% (100)	37% (100)	0.91
Inflammation of the oral mucosa	13% (21)	10% (18)	19% (34)	0.81	23% (57)	18% (49)	0.55

**Table 5 children-08-01173-t005:** Stressors among the respondents.

Stressors	Age			*p*	Sex		*p*
	10–12	13–15	16–18		Girls	Boys	
Characteristics *n*/%							
parents’ finance	10% (16)	7% (13)	21% (37)	0.91	22% (55)	17% (46)	0.91
body image change	83% (134)	80% (146)	86% (152)	0.41	93% (232)	86% (232)	0.12
lack of control over life	26% (42)	55% (100)	58% (103)	0.81	56% (140)	58% (157)	0.44
educational plans	17% (27)	26% (47)	38% (67)	0.01	40% (100)	36% (97)	0.55
plans for the future	18% (29)	27% (49)	29% (51)	0.01	30% (75)	27% (73)	0.19
family relationships	18% (29)	7% (13)	11% (20)	0.41	18% (45)	7% (19)	0.91
relationships with peers	44% (71)	40% (73)	58% (103)	0.01	43% (107)	40% (108)	0.55

**Table 6 children-08-01173-t006:** Patients’ needs.

Needs	High Needs/Moderate Needs %			95% Cl (%)
	10–12	13–15	16–18	
Social needs				
emotional support from society	93	55	59	54–95
support through prayer	90	73	70	69–93
financial assistance	85	64	55	50–91
material support	40	35	29	25–41
the need for contacts with peers	45	30	38	36–48
Caring needs				
improvement of conditions in the hospital	31	30	29	21–30
home treatment support	39	32	23	20–41
better quality of care	24	28	44	20–46
supports from nurses and doctors	40	36	25	22–40
financial support for treatment and rehabilitation	19	21	30	18–33
Educational need				
better cancer education	25	31	39	20–42
access to hospital education	57	59	67	51–72
support in further education	44	41	66	41–72
Psychological needs				
need for love	73	54	43	42–77
the need for compassion on the part of nurses, doctors	88	47	28	25–90
the need for contact with loved ones	93	80	72	69–97
Physical needs				
help with everyday activities	85	31	25	20–87

## Data Availability

Data are available on request due to restrictions of privacy and ethics.

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
