# Peer review of "Problems, Stressors and Needs of Children and Adolescents with Cancer"

_children, 2021, doi:10.3390/children8121173_

Round 1

Reviewer 1 Report

I appreciate this manuscript.
In my opinion, overall, this paper is well-written, well-organized and well-illustrated.
I have only few concerns:
1) why authors initially speack about a clinical interview,  and then present a questionnaire?
2) I believe that it is necessary to report several phrases example comes from the sample when authors present their results.

Author Response

Dear Reviewer,

I would like to thank the reviewer for the valuable comments. After analysing all the comments, I made the following changes:

  1. The "Clinical interview" part was standardized in the article (line 138-143).
  2. The phrases from the sample were quoted in the presentation of the results (line 287-410).

I hope that the changes made are satisfactory and this will allow publication.

Sincerely

Anna Lewandowska

Reviewer 2 Report

This study aims to analyze the problems, stressors and needs of children and adolescents hospitalized due to cancer. I found this paper original since the topic has not been researched as much. It will be interesting to explore a more clinical approach with a more structured questionnaire or a more in depth exploration of how these patients are doing with the biopsychosocial sphere of need and stressors among children and adolescents treated for cancer.

The text is clear and easy to read and the although conclusions are limited with all the evidence and arguments presented which are biopsychosocial sphere of need and stressors among children and adolescents treated for cancer. They address the main question with introductory evidence due the limited data, but it will be interesting to see the questionnaire that was imparted to the patients. 

In conclusion, for future research purposes will be riveting to construct a more structured survey in order to get into a more international population and add more data to this matter. That is an interesting approach to search more un depth how children and adolescents face this stressful disease and help them to be resilient. 

Author Response

Dear Reviewer,

I would like to thank the reviewer for the valuable comments. Our research gave us the opportunity to have a closer look at the problems and needs of children with cancer, and also showed us the shortcomings of the interview. Our interview will be detailed and expanded. In addition to Poland, we will conduct our next research in Slovakia, the Czech Republic and Hungary and compare the results.

Sincerely

Anna Lewandowska